# Role of Simulated *Nannochloropsis gaditana* Digests in Shaping Gut Microbiota and Short-Chain Fatty Acid Levels

**DOI:** 10.3390/nu17010099

**Published:** 2024-12-30

**Authors:** Samuel Paterson, Marta Majchrzak, Dulcenombre Gómez-Garre, Adriana Ortega-Hernández, Silvia Sánchez-González, Miguel Ángel de la Fuente, Pilar Gómez-Cortés, Blanca Hernández-Ledesma

**Affiliations:** 1Institute of Food Science Research (CIAL, CSIC-UAM), Nicolás Cabrera 9, 28049 Madrid, Spain; samuel.paterson@csic.es (S.P.); martamaj11@gmail.com (M.M.); mafl@if.csic.es (M.Á.d.l.F.); 2Microbiota and Vascular Biology Laboratory, Hospital Clínico San Carlos-Instituto de Investigación Sanitaria San Carlos (IdISSC), Prof. Martín Largos, s/n, 28040 Madrid, Spain; mgomezgarre@salud.madrid.org (D.G.-G.); a.ortega.hernandez@hotmail.com (A.O.-H.); silsangon@outlook.es (S.S.-G.); 3Biomedical Research Networking Center in Cardiovascular Diseases (CIBERCV), Monforte de Lemos 3-5, 28029 Madrid, Spain; 4Physiology Department, Faculty of Medicine, Complutense University of Madrid (UCM), Plaza Ramón y Cajal s/n, 28040 Madrid, Spain

**Keywords:** *Nannochloropsis gaditana*, inulin, in vitro gastrointestinal digestion, colonic fermentation, metagenomic analysis

## Abstract

The connection between gut microbiota and factors like diet is crucial for maintaining intestinal balance, which in turn impacts the host’s overall health. *Nannochloropsis gaditana* microalgae is a sustainable source of bioactive compounds, mainly known for its used in aquaculture and extraction of bioactive lipids, with potential health benefits whose effects on human gut microbiota are still unknown. Therefore, the goal of this work was to assess the impact of *N. gaditana* on human gut microbiota composition and derived metabolites by combining the INFOGEST protocol and in vitro colonic fermentation process to evaluate potential effects on human gut microbiota conformation through 16S rRNA gene sequencing and its metabolic functionality. The results have demonstrated the ability of the digests from *N. gaditana* to significantly modify gut microbiota composition, promoting an increase in beneficial bacterial genera such as *Akkermansia*, *Butyricicoccus*, *Eisenbergiella*, *Lachnoclostridium*, and *Marvinbryantia*, in contrast to inulin, after 48 h of colonic fermentation. Additionally, the digests increased the levels of both major and minor short-chain fatty acids (SCFAs), particularly butyric and valeric acids, considered as intestinal biomarkers, and increased ammonium production. This research has demonstrated, for the first time, the potential of *N. gaditana* microalgae as a sustainable agent for influencing the composition and functionality of human gut microbiota.

## 1. Introduction

Nowadays, the relationship between human health and gut microbiota is considered a key target in the study and treatment of many diseases [1]. The community of thousands of highly diverse microorganisms that live throughout the entire gastrointestinal tract has shown to be involved in numerous trophic functions by being a regulatory element of epithelial cell proliferation and differentiation, enhancing the intestinal mucosal barrier and collaborating with the immune system in the protection against antigens and pathogenic germs, among others [2]. In addition, one of its most attractive roles is the regulation of metabolic and nutritional functions such as the fermentation of indigestible fiber from the diet, the production of short-chain fatty acids (SCFAs) and vitamins, and the absorption of ions, which makes the metabolites and soluble factors resulting from its activity have a special interest for human health [3]. For example, the natural carbohydrate inulin from chicory and other plants acts as dietary fiber and needs the gut microbiota microorganisms to break it down, generating multiple benefits for the host health [4]. Moreover, its multifunctionality and complex nature has expanded the number of factors that could be involved in maintaining its homeostasis with the host, as well as the number of strategies that attempt to modulate or influence its behavior [5]. In fact, a dysbiosis state or imbalances in its composition have been hypothesized to be contributing factors in the development of chronic inflammatory diseases such as ulcerative colitis, Crohn’s disease, and obesity, among others [6]. Therefore, the interaction between gut microbiota and its influencing factors, such as diet, plays an important role to maintain intestinal homeostasis, which ultimately affects the host’s health.

The current interest of consumers in sustainable and health-beneficial products, as well as the growth of the world’s population and the impact of the food industry on environmental resources, has led global health organizations such as the Food and Agriculture Organization (FAO) and the World Health Organization (WHO) to recommend a transition of the existing eating pattern to one more focused on the consumption of sustainable functional foods. This circumstance is prompting an ongoing worldwide search for alternative sources of bioactive compounds [7]. Among the different options being evaluated, microalgae stand out due to their sustainable nature and associated health benefits attributed to the presence of potential bioactive compounds capable of exhibiting antioxidant, antitumor, regenerative, antihypertensive, neuroprotective, and immunostimulant effects [8,9]. The consumption of some microalgae species such as *Arthrospira platensis*, *Chlorella vulgaris*, and *Tetraselmis chuii* as novel foods themselves or as food ingredients has been already approved by the European Food Safety Authority (EFSA) and the United States Department of Agriculture (USDA), where the average amount of microalgae incorporated into functional foods typically ranges from 0.5% to 3% of the product’s weight [10]. However, there are other species like *Nannochloropsis gaditana* that are not yet approved, but their promising potential makes them a new niche to explore.

*N. gaditana* is a green, unicellular, subspherical, marine microalgae that has been mainly used for wastewater treatment, animal feeding, aquaculture, and the extraction of bioactive lipids, mostly omega-3 polyunsaturated fatty acids (PUFAs) [11]. Nevertheless, studies focused on the health benefits and potential use of *N. gaditana* biomass as a functional food itself or its addition into known or new functional foods are still scarce [12,13]. To date, the studies that demonstrate the prebiotic and postbiotic capacity of microalgae are limited to the effects of oligosaccharides isolated from these organisms on the growth of specific probiotic cultures. In the study carried out by de Medeiros et al., 2021 [14], the digests of different microalga species showed selective prebiotic effects, stimulating the proliferation *of Lactobacillus-Enterococcus* spp. and *Bifidobacterium* spp. and inhibiting the growth of Prevotellaceae-Bacteroidaceae, *Clostridium histolyticum*, *Eubacterium rectale*, and *Clostridium coccoides*. A more recent study carried out by these researchers demonstrated the stimulating effect of *A. platensis* on the growth and metabolic activity of *Lb. acidophilus* to a greater extent than that exerted by traditional fructooligosaccharides [15]. Additionally, a pre-clinical study with *Phaeodactylum tricornutum* as a suitable diet for mouse feed showed positive effects on microbiota composition and SCFA production, suggesting beneficial effects on gut health [16].

In the case of *Nannochloropsis* sp., it has demonstrated that the fecal microbiota composition of dogs fed with *N. oculata*-supplemented diets was modified by promoting the beneficial *Turicibacter* and *Peptococcus* genera associated with gut health and activation of the immune system [17]. Furthermore, research on European seabass (*Dicentrarchus labrax*) demonstrated that the microalga *N. oceanica* could modulate gut bacterial communities, with beneficial effects on gut homeostasis when combined with other algae like *Gracilaria gracilis* [18]. Specifically, regarding *N. gaditana*, inclusion of its hydrolysate in the *Sparus aurata* fish diet resulted in a beneficial effect on intestinal microbiota beyond its nutritional role without causing significant alterations in intestinal morphology or function [19,20]. Although scarce, these results suggest the potential of this microalga and its metabolites to modulate the gut microbiota. Therefore, the main goal of our work was to assess the role of *N. gaditana*-simulated gastrointestinal digests on human gut microbiota composition and derived metabolites to more comprehensively explore the potential of this microalga as a sustainable modulator agent of human gut microbiota.

## 2. Materials and Methods

### 2.1. Samples and Reagents

Commercial microalga *N. gaditana* biomass was kindly supplied by AlgaEnergy S.A. (Madrid, Spain). Pepsin (EC 232-629-3; 3200 units/mg protein), pancreatin (232-468-9; 8X USP), monosodium phosphate (NaH_2_PO_4_), disodium phosphate (Na_2_HPO_4_), monopotassium phosphate (KH_2_PO_4_), calcium chloride (CaCl_2_), hydrochloric acid (HCl), sodium hydroxide (NaOH), sodium chloride (NaCl), yeast extract, dipotassium phosphate (K_2_HPO_4_), sodium bicarbonate (NaHCO_3_), magnesium sulfate heptahydrate (MgSO_4_·7H_2_O), calcium chloride hexahydrate (CaCl_2_·6H_2_O), Tween-80, hemin, vitamin K, L-cysteine, phosphoric acid (H_3_PO_4_), and inulin from chicory (food grade) were purchased from Sigma-Aldrich (St. Louis, MO, USA).

### 2.2. Simulated Gastrointestinal Digestion

The simulated gastrointestinal digestion of microalgae biomass was conducted following the static in vitro simulation of the gastrointestinal digestion through the INFOGEST 2.0 protocol [21], with some modifications. A pool of saliva was obtained from 7 healthy volunteers, and the simulated gastrointestinal fluid (SGF) and simulated intestinal fluid (SIF) were prepared according to the protocol [21] and preheated at 37 °C. Pepsin from porcine gastric mucosa was mixed with ultrapure water to achieve a final concentration of 33 mg/mL. Pancreatin from porcine pancreas was prepared by dissolving it in SIF to a final concentration of 111.1 mg/mL. Finally, bovine bile was mixed with SIF to a final concentration of 12.9 mg/mL. For the recreation of the oral phase, 1 g of dried microalga biomass was dissolved in 500 µL of Milli-Q water, and 4.5 mL of saliva were added. The mixture was then incubated at 37 °C with 120 rpm of agitation for 2 min in the Environmental Shaker—Incubator ES 20/60 (Biosan Medical-biological Research & Technologies, Warren, MI, USA). After completing the oral phase, 4.8 mL of SGF was added to the bolus, adjusting the pH to 3.0 with 1 M HCl. Next, 3 µL of 0.3 M CaCl_2_ solution and 300 µL of pepsin (ratio enzyme–substrate, E:S, 1:100, *w*/*w*) were added to reach a final volume of 12 mL, and the mixture was incubated at 37 °C with agitation for 2 h. Moreover, the orogastric chyme was mixed with 5.1 mL of SIF, and the pH was adjusted to 7.0 with 1 M NaOH. Then, 1.5 mL of bile (1:50, *w*/*w*), 24 µL of 0.3 M CaCl_2_ solution, and 3 mL of pancreatin (E:S 1:3, *w*/*w*) were added. The resultant homogenate was incubated at 37 °C with agitation for 2 h. At the end of this phase, the enzymes were inactivated at 95 °C for 15 min in a Memmert thermostatized bath (Schwabach, Germany), and the digest was then centrifuged in an EppendorfTM Centrifuge 5804R (Hamburg, Germany) at 2000× *g* for 30 min at 4 °C. Two fractions were obtained, the supernatant corresponding to the absorbable fraction that was discarded and the precipitate corresponding to the non-absorbable fraction (NAF) that was collected. Five digestion replicates were conducted, and the NAFs obtained were pooled and stored at −20 °C until the colonic fermentation assays were carried out in triplicate. A total of 1 g of inulin, a well-known fermentable natural carbohydrate with positive prebiotic effects on the host intestinal health and used to compare the effects of other microalgae species on gut microbiota [22,23], was also digested in quintuplicate, and used as control. Since during the centrifugation of inulin digests, we could lose most of the inulin fraction, we decided to avoid the centrifugation step for this sample and use the complete inulin digest during the colonic fermentation phase to assure the presence of a prebiotic as control along analysis.

### 2.3. Calculation of the Bioaccessibility of Microalgae

Bioaccessibility of *N. gaditana* biomass was measured to investigate the weight changes before and after in vitro digestion and to determine how much *N. gaditana* reaches the colon undigested in the upper gastrointestinal tract using the Jin et al. (2020) formula [23]. In the case of inulin, since the centrifugation step was avoided, the bioaccessibility was not calculated.
(1)$$Bioaccesibility\left(\frac{g}{g}\right)=1-\frac{Nonabsorbable\;fraction\;weight\;(g)}{microalga\;biomassweight\;(g)}$$

### 2.4. Static Colonic Fermentation

For the static colonic fermentation, the fecal inoculum was obtained by combining, in a Stomacher 400 Circulator (Seward, Inc., Bohemia, NY, USA), 1 g of feces (from a healthy volunteer with a Bristol scale value of 4 and a regular and varied diet with no dietetic restrictions) with 10 mL of 0.1 M PBS (pH 7.0) for 5 min to prepare the fecal slurry [24]. Then, 3 g of the NAF, 6 mL of the fecal inoculum, and 54 mL of colon nutrient medium [peptone water (2 g/L), yeast extract (2 g/L), NaCl (0.1 g/L), K_2_HPO_4_ (0.04 g/L), KH_2_PO_4_ (0.04 g/L), NaHCO_3_ (2 g/L), MgSO_4_·7H_2_O (0.01 g/L), CaCl_2_·6H_2_O (0.01 g/L), Tween 80 (2 mL/L), hemin (0.05 g/L), vitamin K (10 μL/L), L-cysteine (0.5 g/L), bile salts (0.5 g/L), and distilled water] were mixed as previously described by Tamargo et al. [25]. The fermentation process was carried out for 72 h in triplicate. Anaerobic conditions were followed at 37 °C, pH 6.8, continuous agitation (120 rpm), and in darkness while taking aliquots at 0, 24, 48, and 72 h. From each 5 mL aliquot, 1 mL was used for microbial counts. The other 4 mL was centrifuged at 10,000 rpm for 10 min at 4 °C using the Eppendorf^TM^ Centrifuge 5804R (Hamburg, Germany), separating the supernatants that were stored at −20 °C for SCFA and ammonium analysis. The pellets were stored at −80 °C until the metagenomic analysis.

### 2.5. Analysis of the Microbial Composition 

#### 2.5.1. Assessment of Microbial Communities

Bacterial counts were performed on general and selective media. Serial dilutions (1/10, 1/100, 1/1000) of the colon digests samples in sterile physiological saline solution (0.9% NaCl) were prepared. Then, spot inoculations (10 μL in triplicate) of each dilution were made on the selected media (BD Difco, Franklin Lakes, NJ, USA): Tryptic Soy Agar (total aerobes); Wilkins Chalgren agar (total anaerobes); MacConkey agar (*Enterobacteriaceae* family); *Enterococcus* agar (*Enterococcus* sp.); De Man, Rogosa, and Sharpe agar (lactic acid bacteria); Tryptose Sulfite Cycloserine agar (*Clostridium* sp.); BBL CHROMAgar (*Staphylococcus* sp.); Bifidobacterium agar modified by Beerens (*Bifidobacterium* sp.); and LAMVAB (*Lactobacillus* sp.). The plates were incubated at 37 °C for 24 to 72 h, depending on the culture medium. All media, except Tryptic Soy Agar and BBL CHROMAgar, were incubated in an anaerobic chamber (BACTRON Anaerobic/Environmental Chamber, Shel Lab, Sheldon Manufacturing Inc., Cornelius, OR, USA). Lastly, the resultant colonies were counted with a colony counter, SC6PLUS (Cole-Parmer, Vernon Hills, IL, USA), and the results were expressed as the logarithm of colony-forming units (CFUs) per mL. The differences in values were considered significant when they were higher or lower than 1 log (CFU/mL) compared to the control (inulin).

#### 2.5.2. Metagenomic Profiling of the Gut Microbiota

##### Extraction and Quantification of Microbial DNA

The isolation and extraction of the microbial DNA from the precipitates obtained at 0 h and 48 h fermentation were carried out using the QIAamp Fast DNA Stool mini kit (Quiagen, Hilden, Germany). DNA integrity was evaluated with the BioAnalyzer 2100 (Agilent, Santa Clara, CA, USA), and the concentration was quantified with the fluorimeter Qubit 3.0 using the dsDNA HS assay (Life Technologies S.A., Alcobendas, Madrid, Spain).

##### 16S Ribosomal RNA Gene Sequencing and Bioinformatics Analysis

All the samples underwent amplification of the 16S ribosomal RNA gene via polymerase chain reaction (PCR) using the Ion 16S Metagenomics kit (Life Technologies S.A.), able to amplify seven hypervariable regions (V2, V3, V4, V6–7, V8, and V9). Then, libraries were generated by repairing the ends of the amplicons with the Ion Plus Fragment Library kit (Life Technologies S.A.) and attaching DNA molecular identifiers using the Ion Express Barcode Kit Adapters (Life Technologies S.A.). Furthermore, libraries were diluted and underwent clonal amplification by emulsion PCR in the Ion OneTouch^TM^ 2 System (Thermo Fisher Scientific, Waltham, MA, USA), and each sample was sequenced using an Ion S5^TM^ System with an Ion 520^TM^ Chip (Life Technologies S.A.).

##### Bioinformatics Analysis

Data from the Ion S5^TM^ System were filtered on the platform-specific pipeline incorporated in the Torrent Suite v5.10 (Thermo Fisher Scientific) to remove low-quality or polyclonal sequences, and to assign the sequences reads to a given barcode. Then, the primers were removed, and the reads were split by 16S region, and trimmed to 165 bp in length using Python. The QIIME 2™ platform [26] was used to assess the quality of the sequencing data, dereplicate the sequences, remove the singletons to obtain representative feature sequences, and group sequences with a 99% identity into Operational Taxonomic Units (OTUs) and align them to the Silva 16S v138 database [27]. Finally, chimeras, mitochondria, and chloroplast were removed using VSEARCH. A single table of OTU abundances from all hypervariable regions per sample was create for statistical analysis using the feature table plugin in the QIIME 2™ platform [26].

To explore α- and β-diversity, rarefaction curves to the minimum sequence depth across samples were created with QIIME 2™. α-diversity was assessed by calculating four metrics: Observed OTUs, Chao1, and Shannon and Simpson indexes. To study β-diversity, the Jaccard dissimilarity index was computed. Principal coordinate analysis (PCoA) was then applied to the resulting dissimilarity matrices, and two-dimensional PCoA plots were used for the visualization of the results.

Taxonomic assignment of the identified OTUs was performed using QIIME 2™ and the SILVA 16S taxonomy database, and a table of the relative abundances of taxa at different levels (phylum, class, order, family, and genus) within each sample was created. Taxa whose relative abundance did not reach 0.01% were filtered out.

The PICURSt2 software package (v2.2.0-b) was used to predict the functional content of microbial communities as Kyoto Encyclopedia of Genes and Genomes (KEGG) ortholog profiles. PICRUSt2 inference relies on 16S rRNA gene sequences and associates OTUs with gene contents. Linear discriminant analysis (LDA) effect size (LEfSe) analysis was used to identify differentially abundant taxa and functional pathways among groups. Taxa with a relative abundance less than 0.01% were filtered for this analysis since they are highly susceptible to statistical noise and mathematical artifacts.

#### 2.5.3. SCFA Analysis

The SCFA analysis was carried out following the García-Villalba et al. [28] methodology. The supernatants were acidified with 0.5% phosphoric acid and mixed with the internal standard (1.97 mM, 2-methylvaleric acid, Sigma-Aldrich). After extracting the mixture with n-butanol, the analysis was performed using a gas chromatograph (GC) equipped with an autoinjector, a DB-WAX capillary column (Agilent Technologies, Palo Alto, CA, USA), and a flame ionization detector. Quantitative data were obtained by calculating the area of each compound relative to the internal standard. The analyses were carried out in triplicate.

#### 2.5.4. Ammonium and Protein Content

The protein concentration of supernatants was carried out using the bicinchoninic acid (BCA) method, using the Pierce BCA kit (Thermo Fisher Scientific), following the manufacturer’s instructions [29]. Bovine serum albumin at concentrations ranging from 50 to 1000 µg/mL was used as standard. The ammonium ion (NH_4_^+^) production was measured using the Photometric Spectroquant^®^ ammonium reagent test (Merck & Co., Kenilworth, NJ, USA), measuring the absorbance at 690 nm using the Biotek SynergyTM HT plate reader (Winooski, VT, USA). A standard curve (2–75 mg NH_4_^+^/L) was used to calculate the content that was expressed as mg NH_4_^+^/L. Both ammonium and protein analyses were carried out in triplicate.

### 2.6. Statistical Analysis

The statistical analysis of data from cultured microbial communities, SCFAs, and protein and ammonium content were performed with GraphPad Prism 8.0 (GraphPad Software, San Diego, CA, USA) using a one-way analysis of variance (ANOVA) followed by a Tukey test. For the metagenomic analysis, differences among groups were analyzed with a Kruskal–Wallis test or a paired *t*-test, as appropriate. The differences between groups regarding the β-diversity were analyzed with a permutation analysis and multiple ANOVA (PERMANOVA) with 999 permutations. Log-fold changes in difference abundance at genus level consists of the application of R package ANCOM-BC2 v.2.4.0 [30] to obtain significantly different effects of *N. gaditana* and inulin. Benjamini–Hochberg false discovery rate correction (FDR) was used to adjust the *p*-value for multiple testing, and FDR < 0.05 was used as a significance threshold. LEfSe consisted of the application of a Kruskal–Wallis test to identify taxa with significantly different relative abundances, followed by a logarithmic Linear discriminant analysis (LDA) score to determine the effect size of each taxon. Finally, the SciPy PYTHON package and IBM^®^ SPSS^®^ Statistics software package version 27 (IBM Inc., Armonk, NY, USA) were used, and for all analyses, differences were considered statistically significant when *p* < 0.05.

A summary of the study workflow is shown in Figure 1.

## 3. Results and Discussion

### 3.1. Bioaccessibility of Microalgae

The bioaccessibility of *N. gaditana* biomass after the simulated gastrointestinal digestion calculated obtained a value of 0.504 g/g of biomass. Although *Nannochloropsis* sp. bioaccessibility has been mainly studied regarding specific compounds such as proteins, carotenoids, or polyunsaturated fatty acids (PUFAs) [31,32], present results were similar to previous works regarding the bioaccessibility of the whole biomass of other thick cell wall, green microalgae like *C. vulgaris*, which obtained a value of 0.40 g/g of biomass [33].

### 3.2. Microbial Composition Analysis

#### 3.2.1. Analysis of Microbial Communities

The effect of the NAFs of *N. gaditana* and inulin digests on the total aerobes and anaerobes microbial counts are shown in Figure 2. In the case of the total aerobes (Figure 2A), an increase was observed at 24 h in both samples, obtaining values of 8.07 CFU/mL and 8.04 CFU/mL for inulin and *N. gaditana*, respectively. After that time, the values measured in the inulin sample remained constant until the end of colonic fermentation (8.25 CFU/mL), while a significant decrease was observed in the microalga sample, reaching 5.34 CFU/mL at 72 h fermentation. Regarding total anaerobes population, both samples experienced an increase throughout the fermentation, and the differences between inulin and microalga samples were not significant (Figure 2B).

In relation to the specific microbial plate counts, the NAF from *N. gaditana* also exerted multiple effects. Thus, the *Enterobacteriaceae* population decreased significantly from 6.42 CFU/mL at 24 h to 4.32 CFU/mL at the end of the fermentation process (Figure 2C). However, the count reached after 24 h fermentation (7.58 CFU/mL) with the NAF from inulin digest remained constant until the end of the fermentation (7.84 CFU/mL). Moreover, the *Staphylococcus* sp. population increased after 24 h (6.20 CFU/mL), when inulin was added to the medium, remaining constant until the fermentation was completed (6.53 CFU/mL) (Figure 2D). Nevertheless, when the NAF from *N. gaditana* digest was added, a significant decrease in the microbial population was observed, decreasing to a value of 5.06 CFU/mL at 72 h.

Differences between the effects of microalga and inulin NAF were also observed for lactic acid bacteria. In *N. gaditana*, there was a slight increase in the microbial population at 24 h (6.61 CFU/mL) that decreased at 48 and 72 h, with values of 5.31 and 5.13 CFU/mL, respectively. However, in the case of inulin, the lactic acid bacteria increased during the fermentation until reaching a final value of 6.29 CFU/mL. No differences were observed for *Enterococcus* sp., *Clostridium* sp., and *Bifidobacterium* sp. microbial counts between *N. gaditana* and inulin at any of the times studied.

The information available on the effects of microalgae *N. gaditana* digests on microbial counts was very limited. Only in the work of Medeiros et al. [14], a significant decrease in the relative abundances of intestinal inflammation-related bacteria like *Eubacterium rectale*, *Clostridium histolyticum*, *Prevotellaceae*, *Bacteroidaceae*, and some *Porphyromonadaceae* was observed in the colonic media when fructooligosaccharides and different microalgae digests were tested, suggesting microalgae’s potential ability to make difficult the growth of inflammation-related bacteria groups like *Enterobacteriaceae* and *Staphylococcus* sp.

The discrepancies found with our results could mainly be related to the microalgae strain, the growing conditions, and/or the experimental design. Furthermore, studies on the growth of *N. gaditana* have shown that both the microalga and its microbiome can adapt to low-oxygen environments [34,35]. During in vitro fermentation, a low-oxygen environment is established, and factors such as the native microbiological load of *N. gaditana*, which could have resisted the digestive process, should not be discarded. Moreover, the phenolic compounds bound to the undigested polysaccharides and proteins in the microalga-NAF could interact with gut microbiota after digestion. This interaction could produce energy that supports primary microbial consumers and their syntrophic partners, also exhibiting the well-known gut-modulating effects of phenolic compounds [36,37,38].

#### 3.2.2. Metagenomic Analysis of Gut Microbiota

The effects of simulated gastrointestinal digests from *N. gaditana* on gut microbiota composition were investigated by assayed bacterial community changes after 48 h of colonic fermentation, using 16S rRNA gene sequencing. Regarding data quality, the total sequence count reached 4,043,573 reads, with an average of approximately 336,964 reads per sample, highlighting the depth of sequencing and the confidence in representing microbial diversity. Sequencing depth varied across samples, ranging from a minimum of 225,476 reads to a maximum of 518,129 reads. This variability provided sufficient coverage across the dataset, ensuring the reliability of the analyses and interpretations. We identified a total of 14,432 OTUs, with 12,001 represented by at least two sequence counts, indicating both high abundance and ecological relevance in the dataset. First, α-diversity, a measurement that considers the internal biodiversity of each sample, considering richness and evenness (Figure 3A–D), was calculated. For this purpose, the sequence depth in all samples was rarefied. As expected, no significant differences were found either in terms of richness (observed OTUs and Chao1) or evenness (Shannon and Simpson indexes) at the baseline sample (zero time) between *N. gaditana* and inulin (Figure 3A–D).

However, after 48 h of fermentation with NAFs from *N. gaditana* and inulin, all α-diversity indexes decreased significantly, indicating a reduction in both bacterial richness and evenness. Previous studies have shown a decline in bacterial abundance during fermentation, likely due to the competitive inhibition of less dominant strains by others [39,40]. Notably, samples treated with *N. gaditana* NAFs exhibited higher observed OTUs and Chao1 indexes compared to inulin-treated samples, though their evenness was similar. This suggests that *N. gaditana* would have a distinct impact on α-diversity compared to inulin.

Regarding β-diversity, which accounts for differences between groups, all samples clustered at the beginning of the fermentation, indicating no initial differences (PERMANOVA, *p* = 0.119) (Figure 4). However, after 48 h of fermentation, both *N. gaditana* and inulin samples displayed changes in bacterial composition relative to the initial time, with significant differences observed between NAF from *N. gaditana* compared to that from inulin (PERMANOVA, *p* = 0.049) (Figure 4). The first two principal coordinates explained over 72% of the variation in bacterial communities (Figure 4). 

Overall, these data would indicate that fermentation of *N. gaditana* had a specific influence on the gut microbiota, consisting in the reduction of α-diversity and alteration of the microbial composition. The influence of *N. gaditana* on the fecal microbiota was then evaluated by analyzing changes in the taxonomic composition of gut microbiota. Both at the beginning of the experiment and at 48 h, Firmicutes, Bacteroidota, and Proteobacteria were the most abundant phyla, accounting for around 95% of bacterial taxa (Figure 5, Appendix A).

After fermentation, *N. gaditana* and inulin showed a notable increase in Proteobacteria, which is consistent with the production of acetic acid that lowers the pH [41]. Regarding the less abundant phyla, there was an increase in the Fusobacteriota and Verrucomicrobiota phyla in samples treated with *N. gaditana* digest at 48 h with respect to 0 h (Fusobacteriota: 6.324 ± 0.684 vs. 0.014 ± 0.010, *p* < 0.01; Verrucomicrobiota: 1.207 ± 0.194 vs. 0.123 ± 0.063, *p* < 0.01), but not in those treated with inulin digest (Fusobacteriota: 0.024 ± 0.007 vs. 0.015 ± 0.009, *p* = NS; Verrucomicrobiota: 0.089 ± 0.078 vs. 0.441 ± 0.125, *p* = NS).

These data, joined to a lower increment in the Proteobacteria phylum, showed a differential effect of *N. gaditana* compared to inulin in the gut microbiota composition. Furthermore, we investigated whether the NAFs from *N. gaditana* were associated with more specific changes in the gut microbiota composition after 48 h of fermentation. Thirty-six genera differed significantly in abundance in comparison with the beginning of the fermentation (FDR-corrected *p* < 0.05) (Figure 6). Twelve genera were significantly increased, and eighteen decreased in both *N. gaditana* and inulin samples (Figure 6, Appendix A). However, NAFs from *N. gaditana*, but not from inulin, induced an increment of the SCFA-producing bacteria *Akkermansia*, *Butyricicoccus*, *Eisenbergiella*, *Lachnoclostridium*, and *Marvinbryantia* (Figure 6, Appendix A).

Two health-related bacteria were identified as key biomarkers in the *N. gaditana* fermentations. A significant increase in *Akkermansia* was observed, which is known to influence host metabolism by producing SCFAs and regulating energy balance, contributing to protective effects against conditions like obesity and diabetes [42]. Additionally, *Akkermansia*’s ability to metabolize mucin may help maintain gut barrier integrity and modulate immune responses. The bacterial genera *Butyricicoccus*, *Eisenbergiella*, *Lachnoclostridium*, and *Marvinbryantia* are key members of the gut microbiota involved in metabolism and inflammation regulation, largely through the production of SCFAs and other metabolites that benefit colonic health and immune modulation. *Butyricicoccus* is a key producer of butyrate, an SCFA that strengthens the gut barrier, reduces inflammation, and supplies energy to colonic epithelial cells [43,44]. *Eisenbergiella* and *Lachnoclostridium* are linked to the fermentation of complex carbohydrates and may indirectly support butyrate production by supplying metabolites that other bacteria can convert into SCFAs [45,46]. *Lachnoclostridium* also participates in the regulation of the immune system. In addition, *Marvinbryantia* is a cellulose-degrading bacterium that produces propionate and acetate, SCFAs known to positively influence glucose and lipid metabolism [47]. Importantly, in human gut microbiota, acetate is the predominant butyrate-producing pathway [48]. Overall, an increment in these genera implies that *N. gaditana* might act as a prebiotic, creating an intestinal environment that promotes SCFA production, which has beneficial effects on gut health and potentially on systemic metabolism.

### 3.3. Microbial Functionality Analysis

In order to study the putative functional profiles of the microbial communities, a LEfSe analysis was performed to compare the KEGG metabolic pathways between *N. gaditana* and inulin after 48 h of colonic fermentation (Figure 7, Appendix A).

*N. gaditana* samples were more enriched in metabolic pathways related to bile acid metabolism (primary and secondary synthesis), sphingolipid metabolism, and α-linolenic acid metabolism, which could be related to the abundance in *Bilophila* and *Akkermansia*, since these bacteria are known to interact with bile acids, suggesting differential regulation of these metabolic pathways [49,50].

In contrast, the inulin samples were more active in pathways related to glycerophospholipid metabolism [51], unsaturated fatty acid biosynthesis, and fatty acid elongation, among others. These samples showed a higher abundance of bacteria such as *Butyricicoccus*, *Clostridium_sensu_stricto*, and *Roseburia*, which are associated with fatty acid metabolism.

#### 3.3.1. SCFA Profile

The concentration of the total, major, and minor SCFAs after *N. gaditana* and inulin NAFs fermentation over time are shown in Figure 8A–H. *N. gaditana* digest was able to significantly increase the total SCFAs concentration up to 46.84 mM at 72 h, surpassing inulin SCFAs production (30.98 mM, Figure 8A). It would be related to the increase in SCFA-producing bacteria from the NAF of *N. gaditana*, such as *Akkermansia*, *Butyricicoccus*, *Eisenbergiella*, *Lachnoclostridium*, and *Marvinbryantia* (Figure 6). The concentrations of the major SCFAs, acetic acid (Figure 8B) and propionic acid (Figure 8C), over time were also higher when colonic medium was treated with NAF from *N. gaditana* digest rather than with inulin, reaching final concentrations of 24.36 and 13 mM, respectively.

Similar results were obtained for butyric acid, whose concentration at the end of the fermentation with the microalga was 2.7-fold higher than that determined when inulin-NAF was added to the medium (Figure 8D).

Several biological functions have been attributed to these major SCFAs, acting as key signaling molecules in human metabolic health [52]. Acetate and propionate have important roles in human metabolism, being utilized in lipid synthesis, acetylation reactions, and as a source of energy [53]. Most obesity animal models and human trials have suggested a beneficial effect of both SCFAs, favoring weight loss and improving insulin sensitivity [54,55,56]. On the other hand, butyric acid stands out as a key metabolite of the colonic microbiome. Recent studies have explored its chemistry, cellular signaling mechanisms, and clinical benefits, particularly for the colonic mucosa, where it serves as the main fuel source for mature colonocytes [57]. Beyond the colon, butyric acid also functions as a local and systemic microbial metabolite with significant anti-inflammatory activity, making it a widely recognized biomarker of intestinal health [58].

Our findings correlate well with previous results from other microalgae species and suggest the potential of the green microalgae *N. gaditana* as an important inducer of gut SCFAs production. Jin et al. [23] demonstrated prebiotic effects of the digests of different microalgae on microbial species involved in propionic acid metabolism, which resulted in an increase in this SCFA. Moreover, de Medeiros et al. [14,15] determined the major SCFAs produced by different microalgae species like *A. platensis*, *C. vulgaris*, and *Desmodesmus. maximus*. Regarding acetic acid, its production was increased when the medium was treated with *C. vulgaris* or *A. platensis* biomass, while butyric acid was highly increased in the fermentation of *D. maximus* biomass after 48 h. A more recent study, focused on *A. platensis* and *P. tricornutum* extracts subjected to in vitro digestion and fermentation, showed beneficial effects because of the increased production of acetic, propionic, and butyric acids [59]. *N. gaditana*-NAF also exerted a stimulatory effect over the production of minor SCFAs when compared to inulin-NAF (Figure 8E–H). A potent stimulating effect was observed for valeric acid production up to 2.74 mM at 72 h fermentation, in contrast to the very low level reached by inulin-NAF (0.05 mM) (Figure 8F). Recently, valeric acid has been identified as one of the most potent histone deacetylase inhibitors, demonstrating anti-cancer, anti-diabetic, antihypertensive, anti-inflammatory, and immunomodulatory activities in different in vitro and in vivo studies [60]. Dietary fiber oat β-glucans, soybean isoflavones, and other plant-derived polysaccharides have been mainly reported to induce the generation of valeric acid by gut microbiota [61,62].

*N. gaditana* is considered one of the microalgae with the thickest cell walls, with polysaccharides mainly composed of glucose (68%), along with rhamnose, mannose, ribose, xylose, fucose, and galactose (4–8%) [63]. Hence, our findings suggested that during simulated gastrointestinal digestion, this polysaccharide-rich cell wall could be partially degraded, being susceptible to the action of gut microbiota and the consequent release of major and minor SCFAs during the fermentation process.

#### 3.3.2. Protein Degradation and Ammonium Production

The protein and ammonium levels throughout the colonic fermentation of the NAF of *N. gaditana* and inulin are shown in Figure 9. *N. gaditana* protein content decreased during the first 48 h of fermentation (Figure 9A), and inversely, the ammonium content increased over time in both samples, being significantly higher in the case of *N. gaditana* (540.91 mg/mL) than that of inulin (200.15 mg/mL) at the end of the fermentation (Figure 9B). Proteins contained in the microalgal biomass, together with peptides and amino acids released during the gastrointestinal digestion, could be a nitrogen source for the growth of gut microbiota, releasing ammonium as a protein metabolism indicator [64]. Ammonium is known to play a significant role in human health by influencing the composition and functionality of the gut microbiota.

The study by Regueiro et al. [65] examined the microbiome’s response to controlled shifts in ammonium and long-chain fatty acids in anaerobic co-digestion systems. They found that high ammonium levels were correlated with shifts in microbial community composition, increasing the abundance of certain bacteria linked to SCFA production, while also leading to volatile fatty acid accumulation. This highlights the complex relationship between ammonium levels and SCFA production, suggesting that the higher SCFA stimulation observed with *N. gaditana*, compared to the inulin control, may also be linked to the increase in ammonium levels.

The main aim of the present study was to assess the role of *N. gaditana* simulated gastro-intestinal digests on human gut microbiota composition, focusing on beneficial bacteria and SCFA production, and present results demonstrated a positive impact. Our findings also suggest its potential to mitigate the growth of harmful microorganisms by increasing beneficial genera such as *Akkermansia*, *Butyricicoccus*, and *Lachnoclostridium*, which could competitively exclude harmful bacteria. On the other hand, enhanced SCFA production, particularly butyric acid, may lower colonic pH, creating a less favorable environment for harmful microorganisms. Additionally, the observed increase in ammonium production might reflect protein fermentation, with its effects depending on the balance with SCFA production. Further studies are needed to clarify the role of *N. gaditana* in modulating harmful microorganisms.

## 4. Conclusions

Our findings support the modulatory role of *N. gaditana* on human gut microbiota composition and derived metabolites after its simulated gastrointestinal digestion. After 48 h of colonic fermentation, NAFs from *N. gaditana* significantly changed gut microbiota bacterial composition, favoring the rise of health-related bacteria genera such as *Akkermansia*, *Butyricicoccus*, *Eisenbergiella*, *Lachnoclostridium*, and *Marvinbryantia*, in contrast with inulin. Moreover, compared to this prebiotic, *N. gaditana*’s simulated gastrointestinal digests increased the production of major and minor SCFAs, especially the key bioactive SCFAs butyric acid and valeric acid, as well as the ammonium production after complete protein digestion. Consequently, this work highlights for the first time the potential of *N. gaditana* microalgae as a new sustainable modulator agent of human gut microbiota composition and functionality. Additional studies should be carried out to elucidate the molecular mechanisms of the compounds responsible for the observed effects. Moreover, the effects of this microalga on the gut microbiota when incorporated into a food matrix should be explored.

## Figures and Tables

**Figure 1 nutrients-17-00099-f001:**
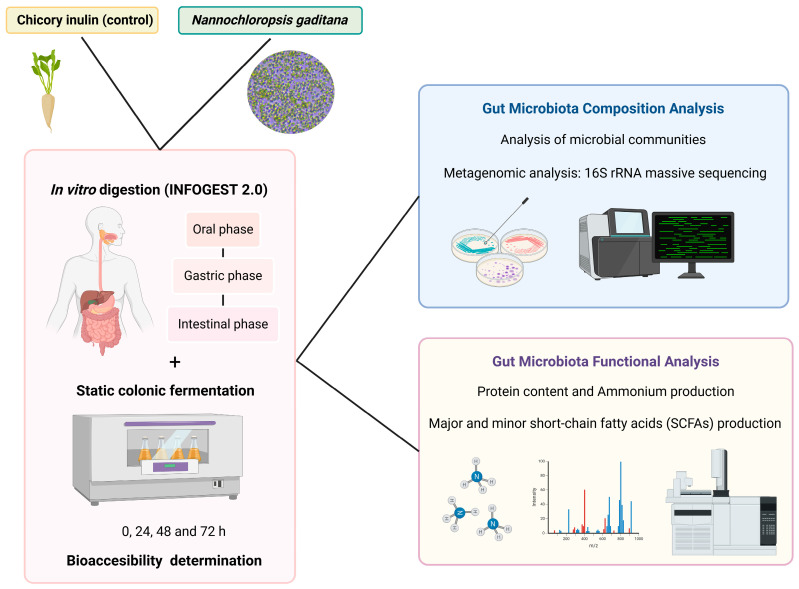
Graphical representation of the study design workflow followed. Pink square: initial static simulation of gastrointestinal digestion, colonic fermentation and bioaccessibility determination of *N. gaditana* and chicory inulin (control). Blue square: gut microbiota microbial composition analysis of *N. gaditana* and chicory inulin (control) simulated digests. Yellow square: gut microbiota SCFA analysis, ammonium and protein determination of *N. gaditana* and chicory inulin (control) simulated digests.

**Figure 2 nutrients-17-00099-f002:**
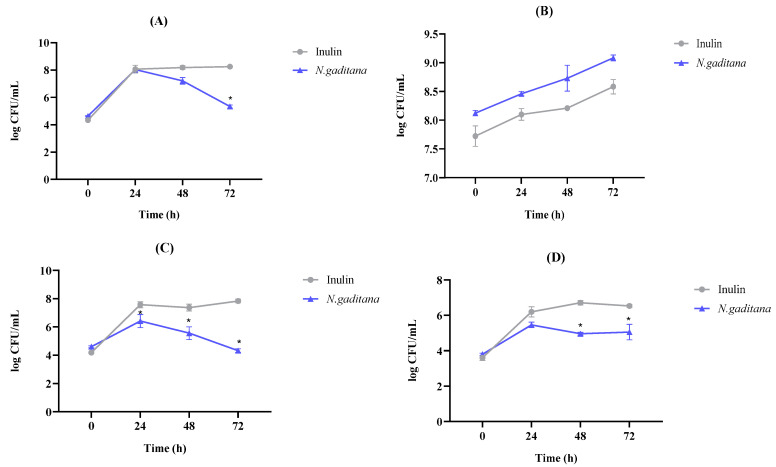
Impact of the non-absorbable fractions (NAFs) of *Nannochloropsis gaditana* and inulin on gut microbiota microbial counts (log Colonies Forming Units (CFUs)/mL) after 0, 24, 48, and 72 h of colonic fermentation. (**A**) Total Aerobes, (**B**) Total Anaerobes, (**C**) *Enterobacteriaceae*, (**D**) *Staphylococcus* sp. *n* = 3 (of a pool of 5 digestions) at each time. * differences in values were considered significant when they were higher or lower than 1 log (CFU/mL) compared to the inulin (control).

**Figure 3 nutrients-17-00099-f003:**
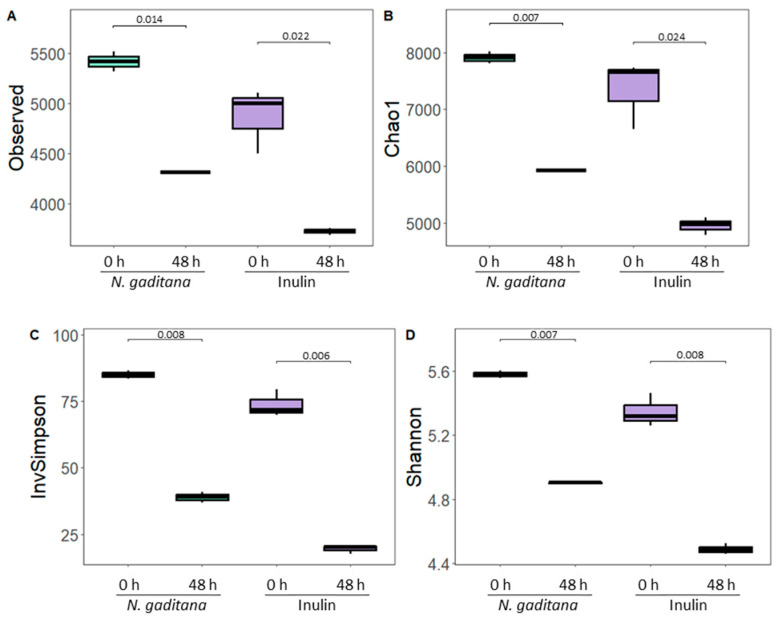
Gut microbiota diversity of the non-absorbable fractions (NAFs) from *Nannochloropsis gaditana* and inulin at zero time and after 48 h. α-diversity is presented by observed Operational Taxonomic Units (OTUs) (**A**), Chao1 (**B**), Simpson richness index (**C**), and Shannon diversity index (**D**). The results are shown in boxplots. *n* = 3 (of a pool of 5 digestions) at each time.

**Figure 4 nutrients-17-00099-f004:**
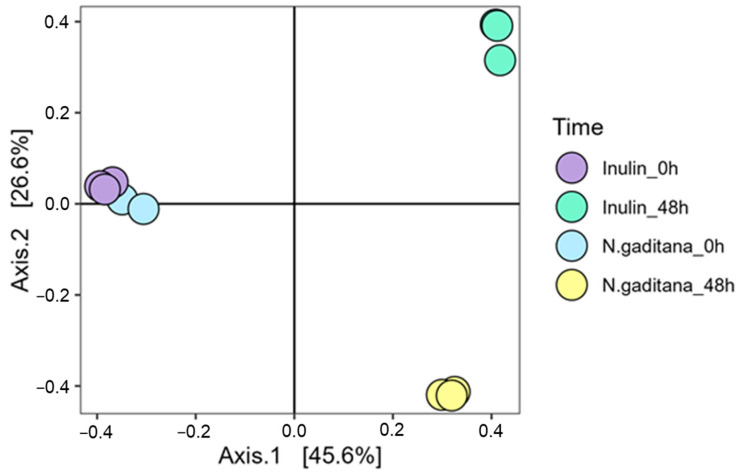
β-diversity presented by PCoA plot of Jaccard dissimilarity index of NAFs from *Nannochloropsis gaditana* and inulin at zero time and after 48 h. PCo1 and PCo2 values for each sample are plotted, with the percentage of explained variance shown in parentheses. *n* = 3 (of a pool of 5 digestions) at each time.

**Figure 5 nutrients-17-00099-f005:**
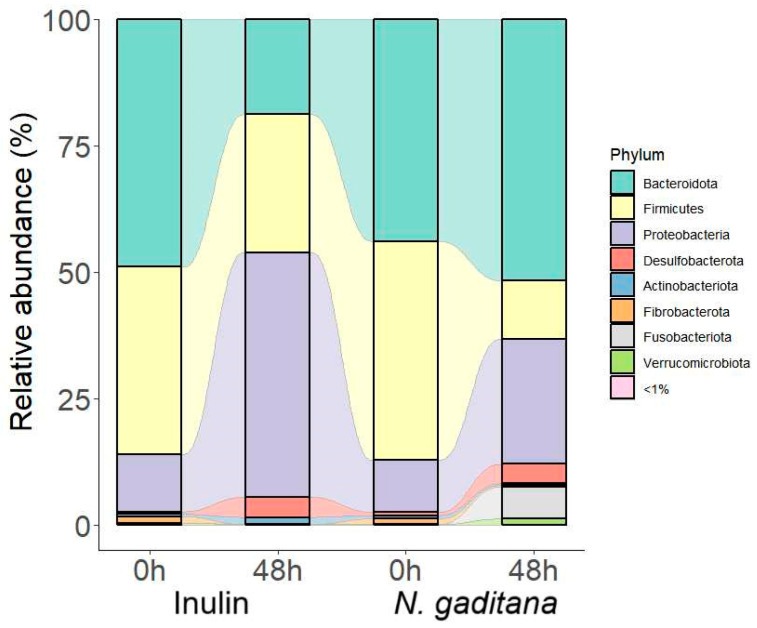
Gut microbiota composition at phylum level of the non-absorbable fractions (NAFs) from *Nannochloropsis gaditana* and inulin at zero time and after 48 h. *n* = 3 (of a pool of 5 digestions) at each time.

**Figure 6 nutrients-17-00099-f006:**
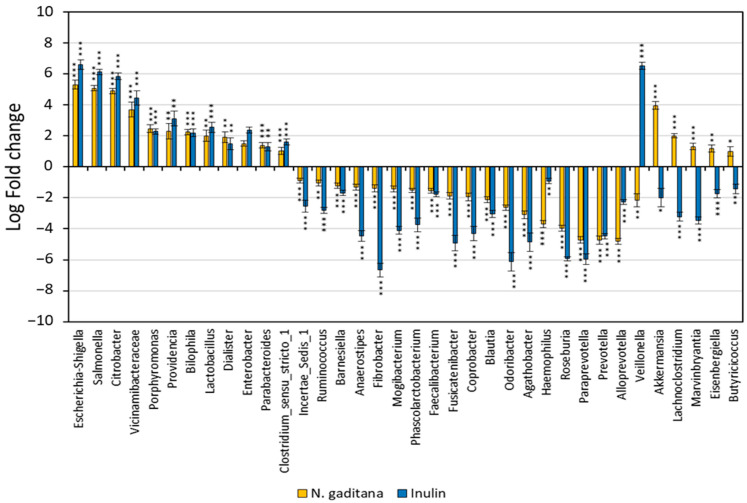
Changes in the abundance of bacteria genera differentially expressed on the non-absorbable fractions (NAFs) from *Nannochloropsis gaditana* and inulin after 48 h using ANCOM-BC. Data are represented by effect size (log-fold change) and standard error bars (two-sided; Bonferroni adjusted) derived from the ANCOM-BC model. *n* = 3 (of a pool of 5 digestions) at each time. * adjusted *p*  <  0.05 vs. zero time, ** adjusted *p* < 0.01 vs. zero time, *** adjusted *p* < 0.001 vs. zero time. Exact adjusted *p* values can be found in Appendix A.

**Figure 7 nutrients-17-00099-f007:**
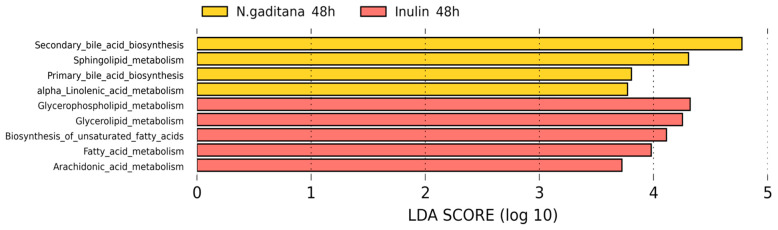
Linear discriminant analysis (LDA) effect size (LEfSe) analysis showing discriminative Kyoto Encyclopedia of Genes and Genomes (KEGG) metabolic pathways between *Nannochloropsis gaditana* and inulin after 48 h of colonic fermentation. All KEGG pathways showed a statistically significant change (*p* < 0.05), with an LDA score threshold set to 2.5.

**Figure 8 nutrients-17-00099-f008:**
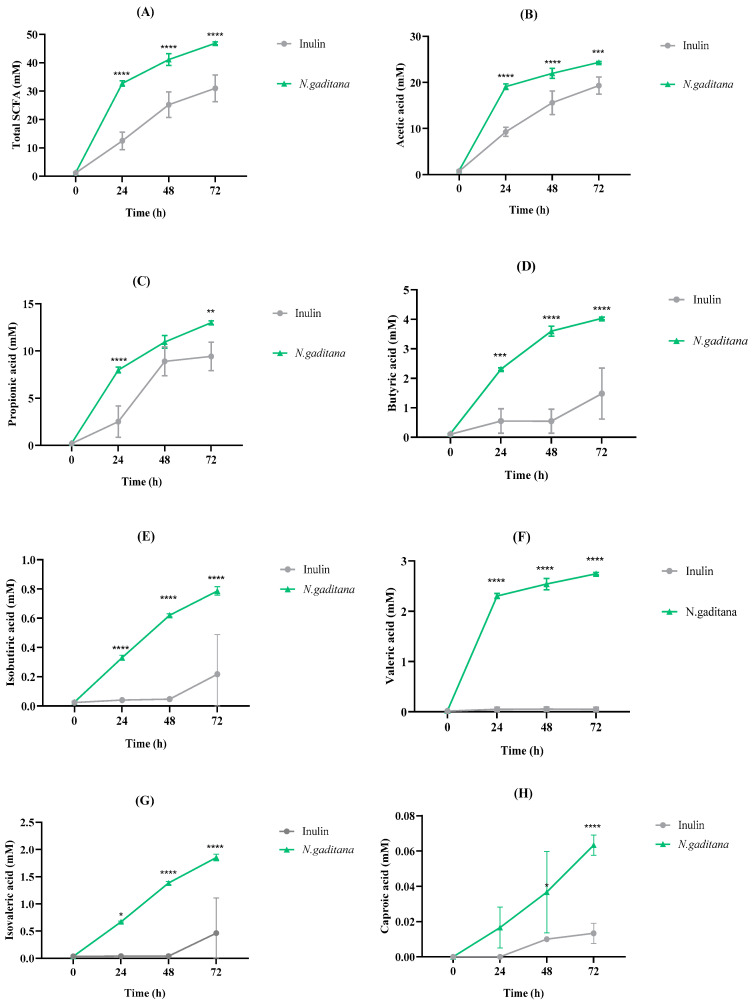
Major and minor short-chain fatty acids (SCFAs) production after the colonic fermentation of the non-absorbable fractions (NAFs) from *Nannochloropsis gaditana* and inulin (control) at 0, 24, 48, and 72 h. (**A**) Total SCFAs, (**B**) Acetic acid, (**C**) Propionic acid, (**D**) Butyric acid, (**E**) Isobutyric acid, (**F**) Valeric acid, (**G**) Isovaleric acid, (**H**) Caproic acid. n = 3 (of a pool of 5 digestions) at each time. Level of significance each time compared to the inulin (control): * 0.01 < *p*-value < 0.05; ** 0.001 < *p*-value < 0.01; *** 0.0001 < *p*-value < 0.001; **** *p*-value < 0.0001.

**Figure 9 nutrients-17-00099-f009:**
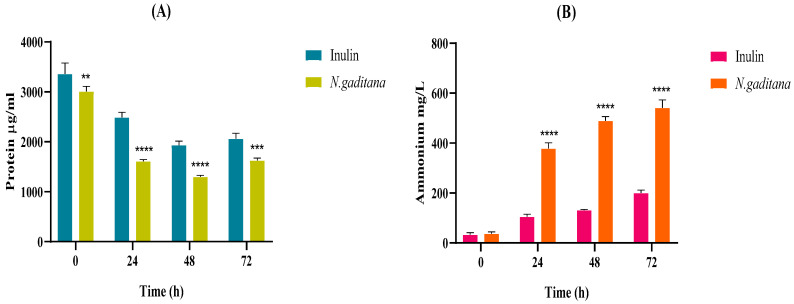
Protein and ammonium levels throughout the colonic fermentation of the non-absorbable fractions (NAFs) of *Nannochloropsis gaditana* and inulin (control) at 0, 24, 48, and 72 h. (**A**) Protein content (µg/mL), (**B**) Ammonium content (mg/L). n = 3 (of a pool of 5 digestions) at each time. Level of significance each time compared to the inulin (control): ** 0.001 < *p*-value < 0.01; *** 0.0001 < *p*-value < 0.001; **** *p*-value < 0.0001.

## Data Availability

The original contributions presented in this study are included in the article and Appendix A. Additionally, raw data and metadata have been deposited with the BioProject database at the National Center for Biotechnology Information (NCBI), accessible at https://www.ncbi.nlm.nih.gov/sra/PRJNA1200596 (accessed on 1 January 2020). Further inquiries can be directed to the corresponding author(s).

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
