# Peer review of "Role of Simulated Nannochloropsis gaditana Digests in Shaping Gut Microbiota and Short-Chain Fatty Acid Levels"

_nutrients, 2024, doi:10.3390/nu17010099_

Round 1
Reviewer 1 Report
Comments and Suggestions for Authors
The authors describe a study that determines the effect of a microalgae, Nannochloropsis gaditana, on the gut microbiota and associated metabolites. An in vitro model was used, and N. gaditana was first put through a simulated upper GI tract digestion. The resultant non-absorbable fraction was then fermented to mimic the colon. The results are reported in this paper.
Here are my comments and suggestions.
· Abstract: The acronym INFOGEST should be defined.
· Introduction: The authors state that N. gaditana could be incorporated into food. Currently, it is used for “wastewater treatment, animal feeding, aquaculture and the extraction of bioactive lipids.” Would it be incorporated into known foods, for example, pasta sauce, or would new foods be made? A sentence or two to provide more information here would be helpful.
· Materials and Methods: The authors should consider a Figure to describe the study design/flow.
· Materials and Methods: How does the dose of N. gaditana relate to what would be in food or taken as a supplement?
· Materials and Methods: A fecal sample from a healthy volunteer was used to inoculate the Stomacher. More information on the subject should be provided, especially since this is an n of one. We know that many components of lifestyle can affect the gut microbiome. For example, what was the Bristol scale score of the sample? What type of diet was the volunteer following?
· Materials and Methods: Was consent obtained for the fecal sample?
· Materials and Methods: Was the fermentation in light or dark? The temperature, pH, and agitation were provided.
· Materials and Methods: How much N. gaditana reaches the colon undigested in the upper GI tract?
· Materials and Methods: Why was inulin selected?
· Results and Discussion: Figure 1 is missing.
· Results and Discussion: The n should be provided on the figures.
· Results and Discussion: Why did gut diversity decrease (Figure 2)?
· Conclusions: The authors conclude that N. gaditana changes the gut microbiota, which their data supports, but go further, stating it would be a “practical food ingredient to enhance health by improving the intestinal microbial environment.” They provide no data to support this statement. The experiment reported in this manuscript did not include mixing N. gaditana into a food matrix, which could significantly reduce any positive effect N gaditana may have. The results suggest that N. gaditana’s effect on the gut microbiome with food should be explored.
Reviewer 2 Report
Comments and Suggestions for Authors
The present study "Role of Simulated Nannochloropsis gaditana Digests in Shaping Gut Microbiota and Short Chain Fatty Acid" by Paterson et al investigated the effects of N. gaditana on human gut microbiota composition and derived metabolites. Overall, the study provided results that indicate N. gaditana could be a potential resource for health beneficial effects, making this work highly significant. However, there are several concerns in this manuscript. The work is mainly based on the sequencing data, therefore detailed information of the raw and processed data should be provided.
Comments
1. What are its impacts on harmful microorganisms?
2. Please add Figure 1.
3. Please include a brief description of the raw and processed reads summary, including the quality information.
4. Did you filter the OTU table for chloroplasts, mitochondria, or samples with low reads? Please add a supplementary with the final OTU table statistics that used for the subsequent analysis.
5. Why is there so much variation in alpha diversity analysis in the 0-hour treatment with N. gaditana and inulin, while the community (beta diversity) is nearly similar?
6. What is the relative abundance of the phyla Fusobacteriota and Verrucomicrobiota that increased with N. gaditana digest but not with inulin digest?
7. How did you analyze and prepare Figure 5? Please add a supplementary table with the relative abundance of the genera at 0 and 48 hours in two different treatment conditions.
8. Why did you remove taxa with a relative abundance of less than 0.01% for the analysis of putative functional profiles? Do they not provide any significant function? The predicted KEGG metabolic pathways should be provided as a supplementary file.
9. For lines 371-385: Please provide the relative abundance of the genus (as a table, heatmap, etc.) and cite it here to support the statement in this paragraph.
10. The raw data needs to be publicly available so that reviewers can check and review it.
Reviewer 3 Report
Comments and Suggestions for Authors
The relationship between gut microbiota and diet is fundamental to maintaining intestinal balance and host health. The microalgae Nannochloropsis gaditana is a source of bioactive compounds described in the literature, including bioactive lipids with expected health benefits, the impact of which on the human intestinal microflora is still unknown. The aim of this study was to assess the impact of N. gaditana and its metabolites on the composition of the human intestinal microflora. In the studies, N.gadianta was compared with the prebiotic - inulin. To assess the impact of N. gaditana, the INFOGEST protocol was used and the composition of metabolites formed during in vitro colonic fermentation was examined to assess the potential impact on the composition of the human intestinal microflora. The 16S rRNA gene was sequenced. The results indicate that N. gaditana metabolites can modify the composition of the intestinal microbiota, promoting the growth of bacterial genera such as Akkermansia, Butyricicoccus, Eisenbergiella, Lachnoclostridium and Marvinbryantia, which are considered desirable. Moreover, the obtained digestion products increased the level of fatty acids, especially butyric acid and valeric acid, while inulin did not have such an effect. Studies indicate that N. gaditana can modulate the composition and physiology of the human intestinal microbiota. Numerous studies have shown that microalgae are a valuable source of protein, amino acids and vitamins of natural origin. It is therefore not surprising that microalgae are used to produce food, animal feed, fertilizers and high-quality bioproducts. Microalgae with unique nutritional values perfectly complement food.
The authors of this paper point to the potential of the microalgae N. gaditana as a new, sustainable factor modulating the composition and functionality of the human intestinal microflora, which could become a practical food ingredient that improves health by improving the intestinal microbial environment. Currently, such "modulators" are very desirable, but one should be careful in drawing conclusions, the authors carried out 5 repetitions in isolated conditions. The authors rightly point out the need to explain the molecular mechanisms of the compounds responsible for the observed effects.The work is well prepared and the procedures are appropriate. It was purposeful to indicate metabolites and describe the impact of microalgae on the composition of microflora. It is not entirely clear why the authors chose inulin. There should be a 2-3 sentence change in the introduction containing information about inulin (inulin is a natural prebiotic - when supplemented regularly, it helps rebuild the intestinal bacterial flora, which contributes to reducing the risk of gastrointestinal infections and increasing the body's immunity, and additionally improving the absorption of calcium from digestive tract. The properties of this substance definitely help control the level of cholesterol and lipids in the blood. Inulin from chicory and other plants acts as dietary fiber - humans it is not digested, and bacteria break it down in the digestive system). It is mentioned in one sentence in lines 135 and 136. There should be information about the use of this reagent in subsection 2.1.
Please correct the caption of chart A in Figure 8.
Round 2
Reviewer 2 Report
Comments and Suggestions for Authors
Thanks for making the necessary changes in the revised manuscript. However, please note that the raw data (mainly the NGS data) must be publicly available before the paper can be accepted for publication. If you have already submitted it, please provide the accession number in the Revised manuscript.
Author Response
Thanks for your comment. Raw data are accessible at https://www.ncbi.nlm.nih.gov/sra/PRJNA1200596. This information has been included in the new version of the manuscript.